# Antimicrobial Resistance of Non-O157 Shiga Toxin-Producing *Escherichia coli* Isolated from Humans and Domestic Animals

**DOI:** 10.3390/antibiotics10010074

**Published:** 2021-01-14

**Authors:** Yanyu Pan, Bin Hu, Xiangning Bai, Xi Yang, Lijiao Cao, Qian Liu, Hui Sun, Juan Li, Ji Zhang, Dong Jin, Yanwen Xiong

**Affiliations:** 1State Key Laboratory of Infectious Disease Prevention and Control, National Institute for Communicable Disease Control and Prevention, Chinese Center for Disease Control and Prevention, Beijing 102206, China; panyanyuicdc@163.com (Y.P.); baixiangning@icdc.cn (X.B.); yangxicdc163@163.com (X.Y.); clj996926@163.com (L.C.); liuqian@email.ncu.edu.cn (Q.L.); sunhui@icdc.cn (H.S.); lijuan@icdc.cn (J.L.); 2Shandong Center for Disease Control and Prevention, Jinan 250014, China; hubinsdcdc@163.com; 3Division of Infectious Diseases, Department of Medicine Huddinge, Karolinska Institutet, 14186 Stockholm, Sweden; 4mEpiLab, New Zealand Food Safety Science & Research Center, Institute of Veterinary, Animal and Biomedical Sciences, Massey University, 4442 Palmerston North, New Zealand; J.Zhang7@massey.ac.nz

**Keywords:** Shiga toxin-producing *Escherichia coli*, STEC infection, antimicrobial drugs, multidrug resistance, whole genome sequencing

## Abstract

Non-O157 Shiga toxin-producing *Escherichia coli* (STEC) is an important pathogen that can cause zoonotic diseases. To investigate the antimicrobial resistance of STEC in China, non-O157 STEC isolates, recovered from domestic animals and humans from 12 provinces, were analyzed using antimicrobial susceptibility testing and whole genome characterization. Out of the 298 isolates tested, 115 strains showed resistance to at least one antimicrobial and 85 strains showed multidrug resistance. The highest resistance rate was to tetracycline (32.6%), followed by nalidixic acid (25.2%) and chloramphenicol and azithromycin (both 18.8%). However, imipenem and meropenem were effective against all isolates. Antimicrobial resistance patterns varied among strains from different sources. Strains from pig, sheep, humans, and cattle showed resistance rates of 100.0%, 46.9%, 30.3%, and 6.3% to one or more antimicrobials, respectively. Forty-three genes related to 11 antimicrobial classes were identified among these strains. The colistin-resistance gene *mcr* was only carried by strains from pigs. A new fosfomycin-resistant gene, *fosA7*, was detected in strains from humans, cattle, and sheep. Whole genome phylogenetic analysis showed that strains from the four sources were genetically diverse and scattered throughout the phylogenetic tree; however, some strains from the same source had a tendency to cluster closely. These results provide a reference to monitor the emergence and spread of multidrug resistant STEC strains among animals and humans. Furthermore, with a better understanding of antimicrobial genotypes and phenotypes among the diverse STEC strains obtained, this study could guide the administration of antimicrobial drugs in STEC infections when necessary.

## 1. Introduction

Shiga toxin-producing *Escherichia coli* (STEC) is an important zoonotic foodborne pathogen, which can cause watery diarrhea, hemorrhagic colitis (HC), and even life-threatening hemolytic uremic syndrome (HUS) [1,2]. STEC O157:H7 was first recognized as a cause of a foodborne outbreak in the USA in 1982, and remains the most predominant and virulent serotype associated with outbreaks and severe human illnesses worldwide [3]. However, in recent years, non-O157 STEC strains have emerged as a major public health concern [4,5]. Sporadic cases or outbreaks caused by non-O157 STEC strains have been reported increasingly, especially strains of several serogroups (O26, O45, O103, O111, O121, and O145, termed as the “top six”) that were the most prevalent during STEC outbreaks [6,7,8].

Shiga toxin (Stx) comprises two immunologically distinct types, Stx1 and Stx2, each of which includes several genetic subtypes. Stx is the essential virulence factor contributing to the development of HUS [9]. Stx acts as ribotoxin that inhibits protein synthesis within sensitive eukaryotic cells and induces apoptosis [10]. To date, the treatment of STEC infections is mainly supportive, including rehydration therapy and dialysis where necessary [11]. The use of antimicrobials to treat STEC infections remains highly controversial and challenging, because some antimicrobial agents can induce Stx production, thus promoting the development of HUS [12,13]. However, other studies have suggested that certain antimicrobials, such as rifaximine, tetracycline, azithromycin, gentamicin, or ampicillin, can block the SOS response (DNA damage response pathway) and Stx production, thus preventing the development of HUS, especially if administered during the early course of the disease [14,15].

STECs have been isolated from a wide variety of domestic and wild animals, while ruminants are recognized as the most important reservoirs of STECs. Human infections are mainly caused by direct contact with infected animals or the consumption of contaminated foods and water [16,17]. As the natural hosts of STECs, cattle, sheep, and pigs are also important repositories for antibiotic resistance genes that potentially impact food and the environment [16,18]. For example, the novel plasmid-mediated polymyxin resistance gene, *mcr-1*, in zoonotic *E. coli* was firstly identified in China in November 2015, and was then detected extensively in various Gram-negative bacterial species from domestic animal hosts, as well as animal-originated food around the world [19,20,21].

Due to the extensive use of antimicrobial agents in farm animals, antimicrobial-resistant bacteria have become a serious issue. A comprehensive understanding of the prevalence of antibiotic-resistant STECs and the mechanisms underlying various resistance patterns could help possible therapeutic options for human infections [11]. However, limited information on antimicrobial resistance (AMR)-STECs is available in China. In this study, we investigated AMR in a diverse collection of non-O157 STEC isolates recovered from cattle, sheep, pigs, and human patients with diarrhea in China. We also screened for resistance genes using whole genome sequencing (WGS) to understand the potential risk of transmission of AMR-STECs from animals to humans. With an advanced understanding of antimicrobial genotypes and phenotypes among diverse STEC strains, the results of our study could guide the administration of antimicrobial drugs in STEC infections when necessary.

## 2. Results

### 2.1. The Distribution of O:H Serotypes and Stx Subtypes

Among 298 non-O157 STEC strains, 271 had draft genomes determined using the Illumina platform, and the remaining 27 had complete genomes sequenced using the PacBio platform. Using the genome sequences, 63 O-serogroups and 26 H-types were identified among the 298 STEC strains, constituting 92 different O:H serotypes. O21:H45 was the most predominant (15.1%, 45/298), followed by O22:H16 and O155:H21 (5.7%, 17/298). The most predominant O-serogroup was O21 (15.1%), followed by O22 (8.7%), O8 (5.7%), O155 (5.7%), and O171 (3.4%). The most predominant O-serogroups from cattle and sheep isolates were O8 (13.8%) and O21 (27.5%), respectively. Non-O157 STEC strains from diarrheal patients belonged to 19 O-serogroups, among which O130 was the most predominant (18.2%), followed by O26 (12.1%) and O117 (9.1%). Although 12 out of 19 O-serogroups were also detected in one or more animal-derived strains, only O8 was identified from all sources (Appendix A).

Two *stx*_1_ and eight *stx*_2_ subtypes were found among the 298 non-O157 STEC strains, respectively, resulting in 14 different *stx*_1_ and/or *stx*_2_ subtype combinations. The most predominant *stx* subtype was *stx*_1a_ (90/298, 30.2%), followed by *stx*_1c_ (58/298, 19.5%) and *stx*_2b_ (34/298, 11.4%). The most predominant *stx* subtype in cattle-origin strains was *stx*_1a_ (20/80), followed by *stx*_1a_ + *stx*_2d_ (13/80) and *stx*_2a_ (11/80). Among strains from sheep, *stx*_1a_ (57/160) was the most predominant, followed by *stx*_1c_ (43/160) and *stx*_2b_ (28/160). All 25 strains from pigs carried the *stx*_2e_ subtype. Six *stx* subtypes or combinations were identified among the human strains, including *stx*_1c_ (15/33), *stx*_1a_ (13/33), *stx*_2e_ (2/33), *stx*_1a_ + *stx*_2b_ (1/33), *stx*_2d_ (1/33), and *stx*_2k_ (1/33). The *stx* subtypes identified in this study are shown in Appendix A.

### 2.2. Antimicrobial Susceptibility

Among the 19 antibiotics tested in this study, the resistance rate toward tetracycline was the highest, with 32.6% (97/298) having resistant strains, followed by nalidixic acid (25.2%), chloramphenicol and azithromycin (18.8%), ampicillin (18.5%), and trimethoprim-sulfamethoxazole (14.1%). All isolates were susceptible to three antibiotics, i.e., ceftazidime-avibactam, imipenem, and meropenem (Figure 1).

Out of 298 non-O157 STEC isolates, 115 were resistant to one or more antibiotics, and isolates from different sources showed different antimicrobial resistance patterns. All 25 isolates from pigs were resistant to one or more antibiotics, among which 96.0% were resistant to tetracycline, 76.0% to trimethoprim-sulfamethoxazole, 72.0% to nalidixic acid, and 32.0% to chloramphenicol. All eight colistin-resistant isolates identified in this study were recovered from pigs. Among the 160 isolates from sheep, 46.9% showed resistance to at least one antibiotic. The resistance rate was highest for tetracycline (36.9%), followed by nalidixic acid and azithromycin (30.0%) and ampicillin (28.1%). Most cattle strains were antimicrobial susceptible, with five strains being resistant to one or more antimicrobials. Among 33 human STEC isolates, 30.3% showed resistance to at least one antibiotic. The resistance rate was highest for tetracycline (30.3%), followed by nalidixic acid (18.2%), azithromycin (12.1%), and trimethoprim-sulfamethoxazole and ampicillin (9.1%) (Table 1 and Appendix A). Isolates resistant to ampicillin, azithromycin, aztreonam, cefotaxime, chloramphenicol, nalidixic acid, tetracycline, and trimethoprim-sulfamethoxazole were found in all sources.

Multidrug resistance (MDR) was identified in 85 isolates (28.5%). The MDR rate was highest among isolates from pigs (21/25, 84.0%,), followed by those from sheep (56/160, 35.0%), humans (5/33, 15.2%), and cattle (3/80, 3.8%). The MDR frequency was statistically different among strains from different sources (*χ*
^2^ = 68.008, *p* < 0.001), especially in the pig-origin strains (Table 1).

### 2.3. Antimicrobial Resistance Genes

In this study, 43 genes associated with resistance to 11 antimicrobial classes were identified (Table 2). The prevalence of resistance genes was highest in the sheep-derived isolates, with 32 AMR-related genes, followed by pig-derived isolates with 29 AMR-related genes. Cattle-derived isolates harbored 23 AMR-related genes, and 17 AMR-related genes were identified in human-derived isolates. The genes related to resistance to four classes of antibiotics (aminoglycosides: *aac(3)-IV*, *aadA16*, *aph(4)-Ia*, *aadA22*; chloramphenicol: *catA2*; quinolones: *qnrB17*; rifampin: *arr-3*) were only found in sheep-derived strains. Four genes involved in two classes of antibiotic resistance (macrolides: *ermB*, *mefB*; colistin: *mcr-1*, *mcr-3.1*) only existed in pig-derived strains. Genes related to chloramphenicol resistance (*catB3*), macrolides resistance (*ermB*) and extended-spectrum β-lactamase (ESBL) resistance (*bla*_CTX-M-15_) were only identified in human-derived strains. Genes implicated in aminoglycoside resistance (*rmtB*) and fosfomycin resistance (*fosA3*) were only found in cattle-derived strains. Twelve genes related to seven antimicrobial resistances were identified in strains from all sources, including aminoglycosides (*aac(3)-IIa*, *aadA5*, *aph(3″)-Ib*, *aph(6)-Id*, and *aadA*), trimethoprim (*dfrA17*), macrolides (*mphA*), quinolones (*qnrS1*), sulfonamides (*sul1*, *sul2*), tetracyclines (*tetA*), and class A β-lactamase (*bla*_TEM-1_).

The aminoglycoside resistance genes were highly diverse, with 18 different combinations being identified in these strains. In cattle isolates, there were three combinations: *aac(3)-IIa* + *aph(3″)-Ib* + *aph(6)-Id* + *ant(3″)-Ia*, *ant(3″)-Ia*, and *aph(3″)-Ib* + *aph(6)-Id* + *ant(3″)-Ia* + *rmtB*. In sheep, pig, and human isolates, there were 10, 13, and 4 different combinations, respectively (Table 2 and Appendix A).

The antibiotic resistance phenotypes corresponded well to the presence of specific antibiotic resistance-related genes. The 50 STEC isolates carrying chloramphenicol resistance-related genes were all resistant to chloramphenicol. Fifty out of 53 STEC isolates carrying macrolide resistance genes were resistant to azithromycin. Eight-nine out of 90 STEC isolates carrying tetracycline resistance genes were resistant to tetracycline (Table 3).

### 2.4. Phylogenetic Analysis

Although non-O157 STEC isolates from the four sources were scattered throughout the phylogenetic tree, they had a tendency to cluster closely. Cluster I only contained sheep-derived isolates, which all belonged to the O21:H25 serotype, carried the *stx*_1a_ subtype, and demonstrated an MDR phenotype. The antibiotic resistance pattern of 89.3% (25/28) isolates in cluster I was AMP-AZM-CHL-NA-TET. These isolates were all from Sichuan province, and they all carried *bla*_TEM-1_. Some other sheep-derived isolates were grouped into cluster IV with the variable presence of AMR genes. Cluster II contained isolates from all four sources, and most of the pig-derived isolates were grouped into this cluster. Six out of eight *mcr* genes were clustered in cluster II and were specific to pig-derived isolates. However, the *tetA* gene was predominant (67.3%) and was shared by isolates from all sources in cluster II. Most of the human-derived isolates were grouped into cluster III, which also contained sheep-derived isolates. Compared with the other clusters, cluster III carried fewer antibiotic resistance genes (Figure 2 and Appendix A).

Strains harboring multiple AMR-related genes were concentrated in clusters I, II, and IV, and strains carrying one or fewer AMR-related genes were concentrated in cluster III or other unnamed clusters. The MDR-strains in cluster I also showed a high frequency of AMR gene combinations (*qnrS1*, *bla*_TEM-1_, *dfrA14*, *floR*, *mphA*, and *tetA*) (Appendix A).

## 3. Discussion

Animals, especially cattle, are important natural reservoirs of STECs. The prevalence and diversity of non-O157 STEC in healthy pigs and cattle in China were reported in previous studies [22,23]. The present study provided an overview of antibiotic resistance of non-O157 STECs from cattle, sheep, pig, and humans in China. These strains are highly diverse, based on serotype, *stx* subtype, and genome analysis. Our results showed that 115 out of 298 strains were resistant to one or more antimicrobials, and MDR was identified in 85 strains. The pig-derived strains showed the highest antibiotic resistance, followed by sheep, human, and cattle strains, which contrasted with other reports from Spain [24] and Mexico [15].

Polymyxin E is considered as the last resort treatment for carbapenem-resistant *Enterobacteriaceae* infection [19]. A recent report showed that the resistance rate toward colistin in pig-derived *E. coli* was about 33.95% [25]. In the present study, colistin-resistant and *mcr*-positive strains were only identified in pig-derived strains. These *mcr*-positive strains were isolated from different regions, including Beijing, Chongqing, and Heilongjiang, demonstrating the wide spread of the *mcr* gene in China.

A new fosfomycin-resistant gene, *fosA7*, was identified in non-O157 STEC isolates from multiple sources (three from clinical patients, three from cattle, and one from sheep) in the present study. This gene was firstly identified in *Salmonella* in 2017, and was considered to reside on the chromosome, which was different from the plasmid-mediated fosfomycin resistance genes (such as *fosA3*) in *E. coli* and other bacteria [26]. We further determined the location of *fosA7* on the STEC chromosome and found that the *fosA7* 5’ end contained a transposase gene (two types, 1254 bp or 1284 bp), which was different from those detected in *Salmonella* (Accession Number: LAOS01000001.1). By BLAST searching against the *E. coli* genome sequences (102,256 genomes in total available from the NCBI nucleotide collection (nr/nt) on 29 June 2020), the *fosA7* gene was also found in 14 other *E. coli* genomes, and 3 out of 14 were from China (Appendix A). This indicated that the fosfomycin resistance gene *fosA7* might spread to *E. coli* in China. In this study, *fosA7*-positive isolates were scattered throughout the phylogenetic tree, indicating the gene might have the potential to transfer between different genetic strains. Fosfomycin could be included in antibiotic sensitivity testing in the future. Notably, the *tetA* gene was predominant and shared by isolates from all sources in phylogenetic cluster II, implying that this cluster might have higher capacity of transfer between different hosts.

Extended-Spectrum β-Lactamases (ESBLs) are one of the main causes of MDR in *Enterobacteriaceae* bacteria [27]. In this study, seven isolates from four provinces in China carried ESBL genes, and all strains showed MDR (Appendix A). Among the class A β-lactamase genes, *bla*_TEM-1_ was the predominant gene (90.4%; 47/52). Notably, most of the *bla*_TEM-1_-positive strains were concentrated in cluster I and cluster III, and most of these strains were isolated in Sichuan province. The co-existence of two AMR genes, *bla*_CTX-M-65_ + *bla*_TEM-1_ and *bla*_CTX-M-15_ + *bla*_TEM-1_, were found. TEM, SHV, and CTX-M are considered as the three main genetic types of ESBLs [28]. CTX-M is the main type in many countries, such as CTX-M-14 in Japan [29] and CTX-M-2 in South America [30]. However, studies have shown wide geographical differences in their distribution [30,31]. We found that the TEM type seems to be dominant in Chinese STECs, which was supported by the results of an early study [32], in which 51 ESBL-producing *E. coli* isolates were found from 912 *E. coli* strains in China. Among them, the TEM type was the second most prevalent type. Notably, our previous study [33] showed that 47 ESBL-producing atypical enteropathogenic *E. coli* (aEPEC) strains were mainly the CTX-M type, and no TEM or SHV types were detected. The reason for this difference requires further analysis.

Some studies have indicated that quinolones, β-lactams, trimethoprim, trimethoprim-sulfamethoxazole, and others can cause bacterial SOS reactions and the massive release of Shiga toxins. However, the influences of different antibiotic classes on Shiga toxin production in vitro differ [34]. Some antibiotics, such as azithromycin, tetracycline, chloramphenicol, and fosfomycin, not only inhibit Shiga toxin release, but also affect the intestinal adhesion of pathogens and have other functions, such that they have been used to treat STEC infection to prevent HUS [15,24,35,36]. We found that the human STEC isolates in this study comprised a high percentage of strains that were susceptible to β-lactams, trimethoprim, and quinolones drugs. It has been reported that when STECs were exposed to inhibitory or sub-inhibitory concentrations of these antibiotics, the SOS reaction would occur [14]. Hence, it is not recommended to use these antibiotics to treat patients with STEC infections. Among these potential antibiotic classes, i.e., tetracycline, chloramphenicol, and azithromycin, high resistance rates were observed. However, all isolates were susceptible to imipenem and meropenem, which might be considered for the treatment of STEC infections when necessary.

## 4. Materials and Methods

### 4.1. Bacterial Strains

A total of 298 non-O157 STEC strains collected during 2009–2019 were used in this study. Strains were isolated from fecal samples from sheep (160), cattle (80), pigs (25), and human diarrheal patients (33) in 12 geographical regions in China (Appendix A). All STEC strains were confirmed by screening for the presence of *stx*_1_ and/or *stx*_2_ genes and biochemical tests as previously described [37].

### 4.2. Antimicrobial Susceptibility Testing

The minimal inhibitory concentrations (MICs) of all isolates were determined by broth microdilution method using the BD Phoenix^TM^ M50 Automated Microbiology System (BD, San Jose, CA, USA). Nineteen antimicrobial agents were tested in this study, including colistin (PB, 0.25–8 μg/mL), chloramphenicol (CHL, 4–32 μg/mL), tetracycline (TET, 1–16 μg/mL), ampicillin (AMP, 2–32 μg/mL), cefoxitin (FOX, 2–64 μg/mL), nalidixic acid (NA, 4–32 μg/mL), ciprofloxacin (CIP, 0.015–2 μg/mL), amikacin (AMK, 4–64 μg/mL), nitrofurantoin (F, 32–256 μg/mL), ampicillin-sulbactam (SAM, 1–32 μg/mL), ceftazidime-avibactam (CZA, 0.25/4–8/4 μg/mL), azithromycin (AZM, 2–64 μg/mL), aztreonam (ATM, 0.25–16 μg/mL), cefotaxime (CTX, 0.25–16 μg/mL), ceftazidime (CAZ, 0.25–16 μg/mL), imipenem (IPM, 0.25–8 μg/mL), meropenem (MEM, 0.125–8 μg/mL), ertapenem (ETP, 0.25–8 μg/mL), and trimethoprim-sulfamethoxazole (SXT, 0.5–8 μg/mL). The qualitative interpretations of susceptible (S), intermediate (I), or resistant (S) strains were determined according to the standard of the Clinical Laboratory Standards Institute guidelines (CLSI 2020). Multidrug resistance (MDR) was defined as resistance to at least one agent in three or more antimicrobial classes [38].

### 4.3. Whole Genome Sequencing (WGS)

Bacterial DNA was extracted using a Wizard Genomic DNA Purification Kit (Promega, Madison, WI, USA) according to the manufacturer’s instructions. Two sequencing libraries were prepared. One was used for sequencing on the Illumina NovaSeq platform with paired-end reads (2 × 350 bp) to obtain the draft genomes (Illumina, San Diego, CA, USA). The adapter and low-quality reads (a quality score below Q20) were removed using fastp 0.20.1, the filtered reads were assembled de novo using SKESA v.2.3.0 [39], and low quality contigs with <500 bp were filtered with Seqkit 0.11.0 [40]. The other library was for sequencing on the PacBio Sequel platform (Pacific Biosciences, Menlo Park, CA, USA). The raw sequencing reads were processed using SMRT Link v5.0.1 (www.pacb.com/support/software-downloads). Briefly, raw reads were trimmed by a “RUN QC” step, and then assembled de novo using a hierarchical genome-assembly process (HGAP), which belongs to a non-hybrid approach and included three steps: Preassembly-Assembly-Consensus polishing [41].

### 4.4. In Silico O:H Serotyping, Stx Subtyping and AMR Gene Screening

The genome sequences were compared to the data at SerotypeFinder (https://cge.cbs.dtu.dk/services/SerotypeFinder/) to determine the O:H serotypes [42]. NCBI AMRFinderPlus [43] and CARD [44] databases were used to screen the antimicrobial resistance genes, using ABRicate version 0.8.10 (https://github.com/tseemann/abricate). A relative coverage threshold of >97%, which was defined as the coverage percentage multiplied by the similarity percentage, was used to define the presence of AMR genes [45]. To determine *stx* subtypes, an in-house *stx* subtyping database was first created using ABRicate by including representative nucleotide sequences of all identified *stx*_1_ and *stx*_2_ subtypes, and then the genome sequences were used to search against the created *stx* subtyping database.

### 4.5. SNP-Based Phylogenetic Analyses

To obtain a high-resolution phylogeny, we adopted a phylogenetic analysis based on single nucleotide polymorphisms (SNPs) [46]. First, the fasta files of the 298 STEC isolates were used to map the sequences to the reference genome Sakai.gbk using Snippy version 4.3.6 (https://github.com/tseemann/snippy) with default parameters. The core alignment file of the SNPs, including invariant sites, was generated, followed by the removal of recombination using Gubbins version 2.3.4 [47]. The non-SNP regions were then removed using SNP-sites [48]. Finally, IQ-TREE version 1.6.8 [49] was used to build the Maximum-Likelihood phylogenetic tree using clean.core.aln (fasta format) as the input file; the branch bootstrap value was estimated 1000 times using the Ultrafast bootstrap algorithm [50].

### 4.6. Statistical Analysis

The distributions of antibiotic resistance in different sources were assessed using the *χ*^2^ and Fisher’s exact tests, using SPSS version 20.0 (IBM Corp., Armonk, NY, USA). *p* < 0.05 was considered statistically significant.

### 4.7. Data Availability

The genome sequences of the 298 non-O157 STEC strains used in this study have been submitted to GenBank. The accession numbers are shown in Appendix A.

## 5. Conclusions

The present study revealed an overview of antimicrobial resistance phenotypes and genotypes among non-O157 STEC strains from diverse sources in China. We identified *mcr-1* and *mcr-3* in pig-derived non-O157 STEC strains, and the existence of the *fosA7* gene in non-O157 STECs from different sources in China. These results suggest that the emergence and spread of multi-drug resistant STECs among diverse animal and human sources should be continuously monitored. In addition, given the challenges associated with the treatment of STEC infections, our study might provide guidance for antimicrobial selection in the clinical treatment of STEC infections.

## Figures and Tables

**Figure 1 antibiotics-10-00074-f001:**
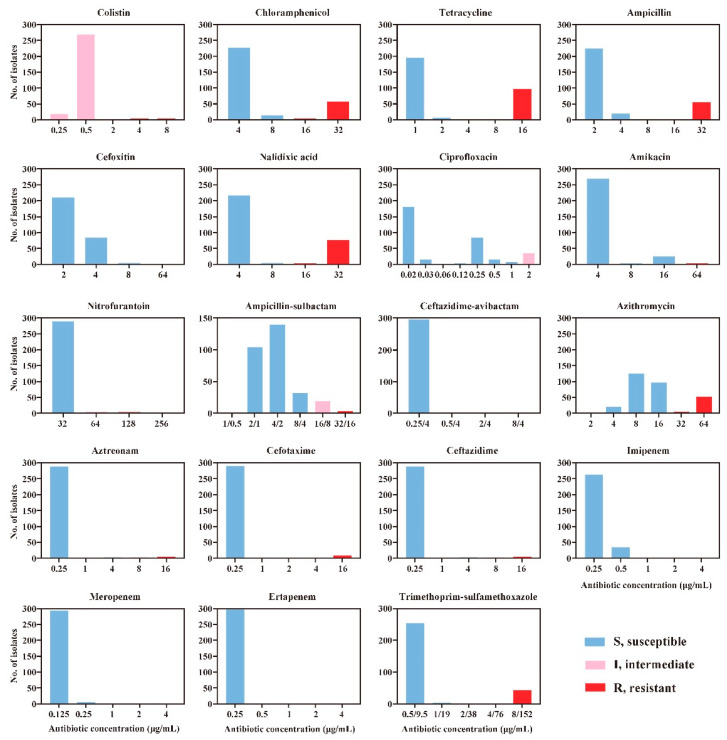
Distribution of minimum inhibitory concentration (MIC) values of 19 antimicrobials among the 298 non-O157 STEC isolates tested in this study.

**Figure 2 antibiotics-10-00074-f002:**
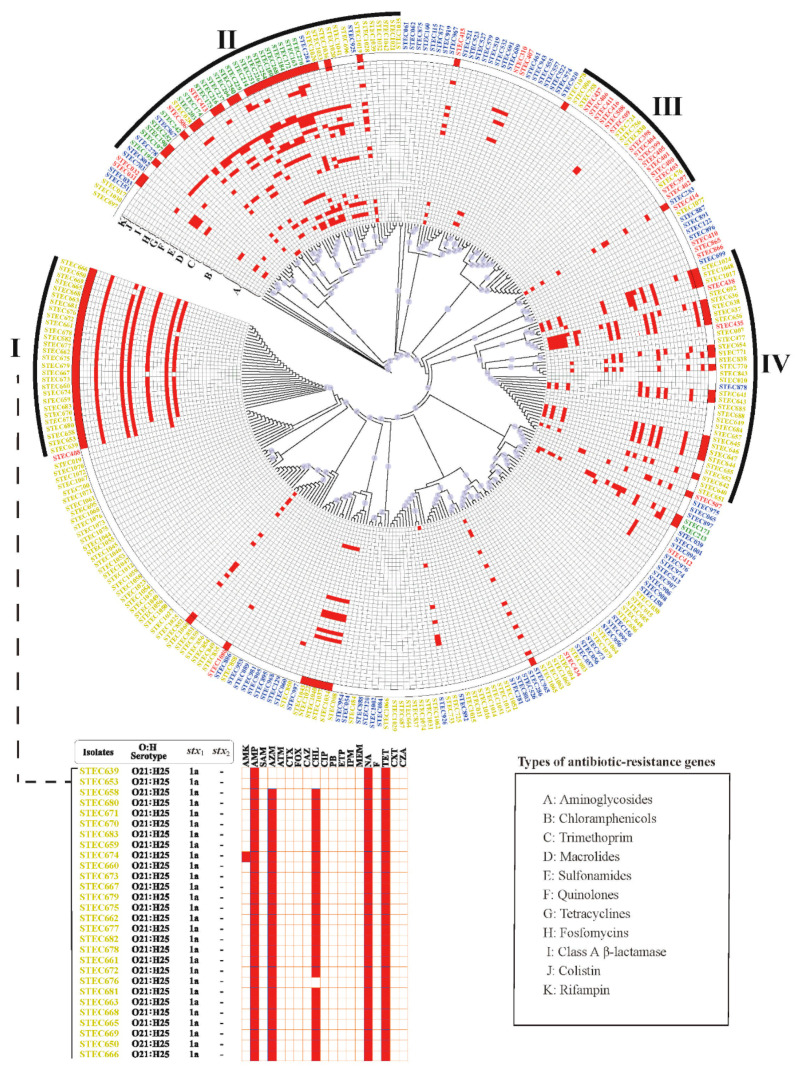
Phylogenetic (cladogram) tree based on core-genome single nucleotide polymorphisms (SNPs) using the Maximum-Likelihood method. The Maximum-Likelihood tree was based on core-genome SNPs of the 298 non-O157 STEC strains from different sources used in this study with strain Sakai (BioSample: SAMN01911278) as a reference. After the removal of recombined regions, a phylogenetic tree was built using 1662 SNP sites by IQ-TREE, which was then visualized and annotated by iTOL. The blue circle on the branch indicates the bootstraps (≥ 60%) of the node. The outer circle indicates the AMR phenotype, with a white background indicating antimicrobial susceptible strains and red indicating MDR. The inner circles (A–K) show the heat map of the AMR genes, red in the white background means the presence of a specific AMR gene. The outermost text indicates the isolate name. Isolates from different sources are indicated using different colors (yellow, sheep; blue, cattle; green, pig; and red, human).

**Table 1 antibiotics-10-00074-t001:** The distribution of multidrug resistance (MDR) among the 298 non-O157 STEC isolates.

No. of Antimicrobial Groups	No. of Resistant Isolates from Different Sources (%)				Total
	Cattle	Sheep	Pig	Human	
0	75 (93.8)	85 (53)	0 (0.0)	23 (69.7)	183 (61.4)
1	1 (1.3)	14 (8.8)	3 (12.0)	2 (6.1)	20 (6.7)
2	1 (1.3)	5 (3)	1 (4.0)	3 (9.1)	10 (3.4)
≥3	3 (3.8) ^a^	56 ^b^ (35.0)	21 (84.0) ^c^	5 (15.2) ^a,b^	85 (28.5)

Note: The same superscript letter (a, b, c) means no significant difference between the two sources.

**Table 2 antibiotics-10-00074-t002:** Antibiotic resistance-related genes among the 298 genome-sequenced non-O157 STEC isolates.

Antibiotic Class	No. of Isolates (%)				
(No. of Isolates)	Resistance Genes	Cattle	Sheep	Pig	Human
Chloramphenicols	*cmlA5*	1 (50.0)	-	-	-
50	*cmlA6*	1 (50.0)	-	4 (66.6)	-
	*floR*	-	37 (90.2)	-	-
	*cmlA5* + *floR*	-	3 (7.3)	-	-
	*cmlA6* + *floR*	-	-	2 (33.3)	-
	*catB3* + *catA1*	-	-	-	1 (100)
Trimethoprim	*dfrA12*	1 (33.3)	1 (2.3)	6 (37.5)	-
50	*dfrA14*	1 (33.3)	36 (83.7)	1 (6.2)	-
	*dfrA17*	1 (33.3)	1 (2.3)	6 (37.5)	2 (66.6)
	*dfrA15*	-	5 (11.6)	2 (12.5)	1 (33.3)
	*dfrA12 + dfrA14*	-	-	1 (6.2)	-
Macrolides	*mphA*	2 (100)	45 (100)	-	3 (100)
53	*mefB*	-	-	2 (66.6)	-
	*ermB* + *mphA*	-	-	1 (33.3)	-
Quinolones	*qnrS1*	1 (100)	48 (87.2)	-	1 (100.0)
66	*qnrB17*	-	5 (9.0)	-	-
	*oqxAB*	-	1 (1.8)	5 (55.5)	-
	*qnrS1* + *oqxAB*	-	1 (1.8)	4 (44.4)	-
Sulfonamides	*sul1*	1 (33.3)	8 (28.5)	4 (21.0)	2 (28.5)
57	*sul2*	1 (33.3)	11 (39.2)	2 (10.5)	3 (42.8)
	*sul3*	1 (33.3)	5 (17.89)	5 (26.3)	-
	*sul1* + *sul2*	-	3 (10.7)	4 (21.0)	2 (28.5)
	*sul1* + *sul3*	-	-	3 (15.7)	-
	*sul2* + *sul3*	-	1 (3.5)	-	-
	*sul1* + *sul2* + *sul3*	-	-	1(5.2)	-
Tetracyclines	*tetA*	3 (100.0)	21 (84.0)	21 (84.0)	7 (77.7)
90	*tetA* + *tetD*	-	4 (16.0)	4 (16.0)	2 (22.2)
Fosfomycins	*fosA3*	1 (25.0)	-	-	-
8	*fosA7*	3 (75.0)	1 (100)	-	3 (100)
class A β-lactamase	*bla* _TEM-1_	1 (50.0)	39 (92.8)	4 (66.6)	1 (50.0)
52	*bla* _CTX-M-55_	1 (50.0)	1 (2.3)	1 (16.6)	-
	*bla* _CTX-M-65_	-	2 (4.7)	-	-
	*bla* _CTX-M-65_	-	-	1 (16.6)	-
	+*bla*_TEM-1_				
	*bla* _CTX-M-15_	-	-	-	1 (50.0)
	+*bla*_TEM-1_				
Colistin	*mcr-1*	-	-	7 (87.5)	-
8	*mcr-3*	-	-	1 (12.5)	-
Rifampin	*arr-2*	1 (100)	7 (58.3)	-	-
13	*arr-3*	-	5 (41.6)	-	-
Aminoglycosides	*ant(3″* *)-Ia*	1 (33.3)	7 (23.3)	4 (21.1)	1 (16.7)
58	*aac(3)-IIa + aph(3″* *)-Ib + aph(6)-Id*	-	10 (33.3)	-	1 (16.7)
	*aph(3″* *)-Ib + aph(6)-Id*	-	4 (13.3)	-	2 (33.3)
	*aph(3″* *)-Ib + aph(3′)-Ia+ aph(6)-Id + ant(3″* *)-Ia*	-	-	6 (31.6)	-
	*aph(3″* *)-Ib + aph(6)-Id + ant(3″* *)-Ia*	-	1 (3.3)	1 (5.3)	2 (33.3)
	*aac(3)-IV + aph(3″* *)-Ib + aph(3′)-Ia + aph(4)-Ia + aph(6)-Id + ant(3″* *)-Ia*	-	4 (13.3)	-	-
	others	2 (66.6)	4 (13.3)	8 (42.1)	-

**Table 3 antibiotics-10-00074-t003:** Correspondence between drug-resistant phenotypes and genotypes of genes associated with resistance to seven classes of antimicrobials.

Antimicrobial Agent	No. of Phenotypic Resistant Isolates		No. of Phenotypic Susceptible Isolates		Sensitivity (%)	Specificity (%)
	Susceptible by Genotype	Resistant by Genotype	Susceptible by Genotype	Resistant by Genotype		
Chloramphenicol	6	50	242	0	89	100
Quinolone *	37	38	195	28	51	87
Aminoglycoside	1	1	239	57	50	81
Macrolide	6	50	239	3	89	99
Tetracycline	8	89	200	1	92	100
Trimethoprim-Sulfamethoxazole	9	33	255	1	79	100
Colistin	2	6	289	1	75	100

* Quinolone results were based on nalidixic acid.

## Data Availability

The data presented in this study are openly available in GenBank and in Appendix A here.

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
