# Peer review of "Antimicrobial Resistance of Non-O157 Shiga Toxin-Producing Escherichia coli Isolated from Humans and Domestic Animals"

_antibiotics, 2021, doi:10.3390/antibiotics10010074_

Round 1
Reviewer 1 Report
The authors present a well-designed study looking at the prevalence of antibiotic resistance genes in STEC isolates from animal reservoirs and human samples. This study is directly comparable to published results from other areas of the world, adding to the surveillance and reporting data about the prevalence of multi-drug resistant isolates around the world. I only have a few questions and formatting comments which are listed below. Overall, this was a very nice manuscript.
Question:
Of the genes queried in the methods, I noticed that there were no data about metallo-beta-lactamases despite their history in other countries in Asia being detected in ESKAPE pathogens. Were there really no MBL genes detected in the genetic studies, or were those genes simply not selected for study? While it may be beyond the scope of this manuscript, it would be interesting to see in future work from the authors. Along the same lines, the prevalence of colistin resistance in the porcine samples is intriguing and would interesting as a future project.
Comments:
Given the number of antibiotic substrates tested, an abbreviation key would be helpful, especially near the MIC data in Figure 1.
In Figure 2, there is some distortion of the box around the key such that the lines are not straight. This should be fixed.
Lines 311-313- there is an inconsistency in the font used which should be fixed so that it matched the rest of the document.
Author Response
Comments and Suggestions for Authors
The authors present a well-designed study looking at the prevalence of antibiotic resistance genes in STEC isolates from animal reservoirs and human samples. This study is directly comparable to published results from other areas of the world, adding to the surveillance and reporting data about the prevalence of multi-drug resistant isolates around the world. I only have a few questions and formatting comments which are listed below. Overall, this was a very nice manuscript.
Question:
Of the genes queried in the methods, I noticed that there were no data about metallo-beta-lactamases despite their history in other countries in Asia being detected in ESKAPE pathogens. Were there really no MBL genes detected in the genetic studies, or were those genes simply not selected for study? While it may be beyond the scope of this manuscript, it would be interesting to see in future work from the authors. Along the same lines, the prevalence of colistin resistance in the porcine samples is intriguing and would interesting as a future project.
Response: Metallo-beta-lactamase genes, such as IMP, VIM, NDM-1, etc., were included in database that we used for screening AMR genes; however, we did not find those genes in the strains included in this study.
As suggested, we will pay more attention to these AMR genes in future surveillance.
Comments:
Given the number of antibiotic substrates tested, an abbreviation key would be helpful, especially near the MIC data in Figure 1.
Response: We have used the full names of each antibiotic substrate in Figure 1. The abbreviations of antibiotic substrates are given in the Materials and Methods section.
In Figure 2, there is some distortion of the box around the key such that the lines are not straight. This should be fixed.
Response: We have fixed the problem and re-uploaded it.
Lines 311-313- there is an inconsistency in the font used which should be fixed so that it matched the rest of the document.
Response: Corrected accordingly.
Again, we sincerely thank all the reviewers for their constructive suggestions. We hope that these modifications will render our paper acceptable for publication in the Antibiotics.
Yours sincerely,
Dr. Yanwen Xiong
State Key Laboratory of Infectious Disease Prevention and Control,
National Institute for Communicable Disease Control and Prevention, China CDC.
Reviewer 2 Report
The manuscript by Pan et al provides a descriptive analysis of non-O157 Shiga
toxin-producing Escherichia coli regarding antimicrobial susceptibility testing, the diversity of stx and resistance gene presence.
The methods used in this manuscript are sound, but the results and discussion need some contextualization in order for this study to gain greater importance, especially given the data already available in the manuscript.
For example, authors put a lot of emphasis on the HUS in the introduction, but this is lost in the rest of the paper. There have been studies that have tried to attest whether specific stx subtypes have been associated with higher chances of HUS. This can be mapped to the phylogenetic tree and try to understand whether specific animal types or clusters in the tree could potentially be associated with these stx. This could be important to identify if there is an animal reservoir in China that is more associated with the onset of this disease in non-O157 E. coli.
The same can be said for the resistance genes. Authors talk about the presence of resistance genes, but not much regarding the four clusters in the phylogenetic tree except for cluster I.
Given that in cluster II there is the mixing of many sources, what resistance genes are also found here? This is important because clearly this cluster might have higher capacity of transfer between different hosts, and the identification of genes that confer resistance to critically important antibiotics in these isolates is of great concern.
Partially, I think this lack of information is due to the representation of the phylogenetic tree. While this current representation gives visually some idea of what the isolates might have because of the presence/absence of resistance genes, the rings do not show which specific gene. Potentially authors can also create a phylogenetic tree where important resistance genes are included, rather than the antimicrobial classes. This can help the text in the results and discussion. For example, are all mcr carrier pig isolates all in the same cluster? Is the same true for the CTX-M-15 human isolates? What is the distribution of the fosA7 gene on the tree? Authors talk about it in the discussion but not in the results. The discussion of this section should also include the geographic origin of the isolates so that one understands whether the presence of certain genes is allocated to specific geographic locations.
Minor comments:
Abstract:
Line 22 – change sensitivity to susceptibility
Lines 23-25 – this sentence is confusing
Lines 27-28 – resistance to what?
Introduction
Lines 47 and 52. Reference is missing
Results
Remove ampC from the results. ampC is an intrinsic to E. coli which is usually silenced and does not provide additional epidemiological information.
Line 139 – ermB is not a chloramphenicol resistance gene.
Line 210 – perhaps include the accession if of Salmonella
Figure 1. Add name of antibiotics in the panels or on the legend
Figure 2. Add the description of the heat map. I imagine that blue in the circles mean that gene is not present. I would rather use white for absence of a gene rather than blue. Also mention that the thickness of the rings represents the number of genes in each class. However, having a description of what each gene represents the ring is more informative for the reader.
Author Response
Comments and Suggestions for Authors
The manuscript by Pan et al provides a descriptive analysis of non-O157 Shiga toxin-producing Escherichia coli regarding antimicrobial susceptibility testing, the diversity of stx and resistance gene presence.
The methods used in this manuscript are sound, but the results and discussion need some contextualization in order for this study to gain greater importance, especially given the data already available in the manuscript.
Response: We agree with the reviewer’s suggestion. We have therefore rephrased the text, especially the results and discussion sections.
For example, authors put a lot of emphasis on the HUS in the introduction, but this is lost in the rest of the paper. There have been studies that have tried to attest whether specific stx subtypes have been associated with higher chances of HUS. This can be mapped to the phylogenetic tree and try to understand whether specific animal types or clusters in the tree could potentially be associated with these stx. This could be important to identify if there is an animal reservoir in China that is more associated with the onset of this disease in non-O157 E. coli.
Response: We thank for the reviewer for their insightful comment. Since none of our strains was from patient with HUS, we did not describe HUS in the rest of the paper (especially the results section). However, the aim of this study is to provide references for the antibiotic use in treatment of STEC infection including HUS, including an advanced understanding of the antibiotic genotypes and phenotypes among a diverse collection of non-O157 STEC strains obtained in this study. Therefore, we think it would make sense to mention HUS, especially its challenging treatment. We have added these brief implications in the introduction section.
Concerning the stx subtypes and its association with HUS, this is a bit out of scope of this study, we did not interpret this in our study. However, we agree with the reviewer’s suggestion that it is an important part. We tried to identify the specific stx subtypes in animal reservoirs associated with human-origin. We add this briefly in the discussion section.
The same can be said for the resistance genes. Authors talk about the presence of resistance genes, but not much regarding the four clusters in the phylogenetic tree except for cluster I.
Given that in cluster II there is the mixing of many sources, what resistance genes are also found here? This is important because clearly this cluster might have higher capacity of transfer between different hosts, and the identification of genes that confer resistance to critically important antibiotics in these isolates is of great concern.
Response: Thank you for this thoughtful comment. According to the phylogenetic tree, Cluster â…¡ contained isolates from all four sources, and most of the pig-derived isolates were grouped into this cluster. Six out of eight mcr genes were clustered in cluster II and were specific to pig-derived isolates. However, the tetA gene was predominant (67.3%) and shared by isolates from all sources in cluster II. We have added the relevant description in revised manuscript.
Partially, I think this lack of information is due to the representation of the phylogenetic tree. While this current representation gives visually some idea of what the isolates might have because of the presence/absence of resistance genes, the rings do not show which specific gene. Potentially authors can also create a phylogenetic tree where important resistance genes are included, rather than the antimicrobial classes. This can help the text in the results and discussion. For example, are all mcr carrier pig isolates all in the same cluster? Is the same true for the CTX-M-15 human isolates? What is the distribution of the fosA7 gene on the tree? Authors talk about it in the discussion but not in the results. The discussion of this section should also include the geographic origin of the isolates so that one understands whether the presence of certain genes is allocated to specific geographic locations.
Response: Thank you for this critical suggestion. We tried to draw a phylogenetic tree with the names of resistance genes initially, but the readability was very poor because of the small loop (thickness) of each resistance gene cause by the huge number of strains and the types of resistance genes. Considering the size of the figure, we used antimicrobial classes on the tree instead of all AMR genes.
We have added some descriptions in the results, including the distribution of special antibiotic resistance genes and geographic information in different clusters. We have also modified the discussion part accordingly.
Minor comments:
Abstract:
Line 22 – change sensitivity to susceptibility
Response: The sentence has been corrected.
Lines 23-25 – this sentence is confusing
Response: The sentence has been rephrased.
Lines 27-28 – resistance to what?
Response: We have change the text to “showed resistance rates of 100.0%, 46.9%, 30.3%, and 6.3% to one or more antimicrobials”.
Introduction
Lines 47 and 52. Reference is missing
Response: We have checked and fixed all the references.
Results
Remove ampC from the results. ampC is an intrinsic to E. coli which is usually silenced and does not provide additional epidemiological information.
Response: We agree with the reviewer’s suggestion. We have deleted all ampC results from the manuscript.
Line 139 – ermB is not a chloramphenicol resistance gene.
Response: This has been corrected.
Line 210 – perhaps include the accession if of Salmonella
Response: We added the accession number (LAOS01000001.1) of Salmonella.
We found that the flanking region of fosA7 contained a transposase gene in STEC. In a previous report, there was no description about the transposase gene of the flanking region of the fosA gene in the Salmonella chromosome. We further checked the Salmonella sequence that reported the fosA7 gene for the first time, and no transposase gene was found in the flanking region of the fosA7 gene. We have added this to the Discussion section.
Figure 1. Add name of antibiotics in the panels or on the legend
Response: We added the full name of each antibiotic in the panels.
Figure 2. Add the description of the heat map. I imagine that blue in the circles mean that gene is not present. I would rather use white for absence of a gene rather than blue. Also mention that the thickness of the rings represents the number of genes in each class. However, having a description of what each gene represents the ring is more informative for the reader.
Response: In figure 2, the outer circle indicates AMR phenotype, white background means susceptible, red means MDR. In the inner circles, we used colored backgrounds to differentiate AMR gene classes (A-K in the panel), red in the different color background means the presence of a specific AMR gene. We added these descriptions to the figure legend.
Again, we sincerely thank all the reviewers for their constructive suggestions. We hope that these modifications will render our paper acceptable for publication in the Antibiotics.
Yours sincerely,
Dr. Yanwen Xiong
State Key Laboratory of Infectious Disease Prevention and Control,
National Institute for Communicable Disease Control and Prevention, China CDC.
Reviewer 3 Report
Antimicrobial resistance of non-O157 Shiga toxin-producing Escherichia coli isolated from human and domestic animals
In this study, non-O157 STEC isolates recovered from domestic animals and humans from twelve provinces in China, were analyzed by antimicrobial sensitivity testing and whole genome characterization. Out of 298 isolates, comprising 92 O:H serotypes and 14 stx1/stx2 subtype combinations, 115 strains showed resistance to at least one antimicrobial, and 85 strains showed multidrug resistance.
Overall, the work is done well, however, the manuscript needs a moderate revision to be reconsidered for publication. Therefore, in my point of view, the manuscript could be considered for acceptance but not in its current form. Having said that following revisions are suggested;
Comments:
- The abstract is descriptive and qualitative. Normally an abstract should state briefly the purpose of the study undertaken and meaningful conclusions based on the obtained results.
- Hence, this needs rewriting. I would expect brief, yet concise, the quantitative data description of the results in the abstract.
- The given list of keywords is superficial with broader terms. More specific terms should be used. Replace accordingly.
- The introduction is short. More literature should be added with recent and relevant literature.
- The novelty of the study should be clearly highlighted in the manuscript at the end of the introduction section, as there are some existing literature reports.
- The level of English used is not up to the journal standard. Throughout the manuscript, the level of English used is not up to the standard of the journal. The sentences are long and badly worded with repetitive words. Please consider breaking longer sentences into smaller fragments for easy understanding. The authors are advised to seek help from a native English speaker.
- The conclusion is superficial. Herein, I would like to see the major findings and how they are addressing the left behind research gaps and covering current challenges.
- Referencing is not right. Literature needs to be updated with care.
- Editorial issues: The Latin names and Greek letters should be presented in italic in the whole manuscript, the unit presentation should be unified in the whole manuscript, abbreviations presentation should be unified.
Author Response
Comments and Suggestions for Authors
Antimicrobial resistance of non-O157 Shiga toxin-producing Escherichia coli isolated from human and domestic animals
In this study, non-O157 STEC isolates recovered from domestic animals and humans from twelve provinces in China, were analyzed by antimicrobial sensitivity testing and whole genome characterization. Out of 298 isolates, comprising 92 O:H serotypes and 14 stx1/stx2 subtype combinations, 115 strains showed resistance to at least one antimicrobial, and 85 strains showed multidrug resistance.
Overall, the work is done well, however, the manuscript needs a moderate revision to be reconsidered for publication. Therefore, in my point of view, the manuscript could be considered for acceptance but not in its current form. Having said that following revisions are suggested;
Comments:
- The abstract is descriptive and qualitative. Normally an abstract should state briefly the purpose of the study undertaken and meaningful conclusions based on the obtained results.
- Hence, this needs rewriting. I would expect brief, yet concise, the quantitative data description of the results in the abstract.
Response: We have revised the abstract accordingly, which included the purpose, the main quantitative findings, and the meaningful conclusions of the study.
- The given list of keywords is superficial with broader terms. More specific terms should be used. Replace accordingly.
Response: The keywords have been replaced accordingly.
- The introduction is short. More literature should be added with recent and relevant literature.
Response: We have expanded the introduction and added more references.
- The novelty of the study should be clearly highlighted in the manuscript at the end of the introduction section, as there are some existing literature reports.
Response: We have highlighted the novelty of this study at the end of introduction.
- The level of English used is not up to the journal standard. Throughout the manuscript, the level of English used is not up to the standard of the journal. The sentences are long and badly worded with repetitive words. Please consider breaking longer sentences into smaller fragments for easy understanding. The authors are advised to seek help from a native English speaker.
Response: We asked Elixigen Corporation (Huntington Beach, California, USA) for helping in proofreading and editing the English of the revised manuscript. The manuscript has now been carefully checked for English grammar and spelling.
- The conclusion is superficial. Herein, I would like to see the major findings and how they are addressing the left behind research gaps and covering current challenges.
Response: We have briefly highlighted the major findings from this study in the conclusion, which could add scientific value to the research filed.
- Referencing is not right. Literature needs to be updated with care.
Response: We have checked and corrected all the references.
- Editorial issues: The Latin names and Greek letters should be presented in italic in the whole manuscript, the unit presentation should be unified in the whole manuscript, abbreviations presentation should be unified.
Response: We have checked and corrected the Latin names, Greek letters, the units, and abbreviations throughout the revised manuscript.
Again, we sincerely thank all the reviewers for their constructive suggestions. We hope that these modifications will render our paper acceptable for publication in the Antibiotics.
Yours sincerely,
Dr. Yanwen Xiong
State Key Laboratory of Infectious Disease Prevention and Control,
National Institute for Communicable Disease Control and Prevention, China CDC.
Round 2
Reviewer 2 Report
The revised version by Pan et al addresses the concerns of the reviewers and the manuscript has more contextualized information that adds value.
However, as with the previous version, the phylogenetic tree still needs some work. Given that there are too many colors, it makes it difficult to follow the figure. I would advise authors to use white squares for absence and then a colored square when the resistance gene for that class is present. This will make picture clearer to interpret. Having too many colors makes it difficult to interpret and also to distinguish. For comparison, look at figure 1 of the following manuscript: https://doi.org/10.1371/journal.pone.0206252
Since it is understandable that authors cannot put all resistance genes in the tree for readability, authors could create a supplementary table with the composition of genes for each isolate and their phylogenetic cluster. This can help readers find further information that is not currently in the text or tables.
Minor comments:
Line 153/238. TEM-1 is not an ESBL but a narrow spectrum ß-lactamase. Please revise the numbers in line 238 and the supplementary table 4. Only CTX-M-15, CTX-M-55 and CTX-M-65 are ESBLs in this study
For consistency, authors should also use one designation for the different aminoglycoside modifying enzymes. For example, in line 151 the gene designation was used, but in the table the capitalized version referring to the protein is used. Since all the other resistance genes are using the gene form (lower case italicized), I would suggest using this designation.
Reviewer 3 Report
Accept
Author Response
Again, we sincerely thank you for your constructive suggestions.